# Characteristic genes and immune landscape of interstitial cystitis

**Junhao Wang[1‡], Yan Zhou[1‡], Jie Hu[2], Jianpeng Han[1], Jianyong Feng[1], Kuo Guo[1], Wenbin Chen[1], Yanrui Yun[1]\*, Yongzhang Li[1]\***

1 Department of Urology, Hebei Provincial Hospital of Traditional Chinese Medicine, Shijiazhuang, Hebei Province, China, 2 Department of Urology, Langfang People's Hospital, Langfang City, Hebei Province, China

‡ Junhao Wang and Yan Zhou are co-first author. They have contributed equally to this article.
* hbzylyz@126.com (YL); 18033709955@163.com (YY)

## Abstract

### Background

Interstitial cystitis (IC) was still a disease with the exclusive diagnosis and lacked an effective gold standard. It was of great significance to find diagnostic markers for IC. Our study was aimed to screen characteristic genes via machine learning algorithms, characterize the immune landscape of IC, and show correlations between characteristic genes and immune cell subtypes.

### Methods

RNA sequencing data sets on IC were downloaded from Gene Expression Omnibus (GEO) database, including GSE57560, GSE11783 and GSE621, whose corresponding platforms were GPL16699, GPL570 and GPL262 respectively. Three machine learning algorithms were applied for identification of characteristic gene for IC. Single sample Gene Set Enrichment Analysis (ssGSEA) was applied to figure out the immune cell infiltration (ICI) of IC and normal tissue samples. Correlation analysis was performed via Spearman test. Receiver operator characteristic curve (ROC) was used to evaluate diagnostic efficacy of key genes.

### Results

CCL18, MMP10 and WIF1 were identified as characteristic gene via machine learning algorithms. MMP10 and CCL18 were with higher expression in IC tissues compared with normal bladder tissues, while WIF1 had lower expression in IC tissues (P < 0.05). These three genes had good diagnostic efficacy for IC. Compared with normal bladder tissues, 18 immune cell subtypes were up-regulated in interstitial cystitis tissues (P < 0.05). MMP10 and CCL18 were positively correlated to immune scores in IC, while WIF1 was negatively correlated to immune scores (P > 0.05).

**Data availability statement:** All relevant data are within the manuscript and its Supporting Information files.

**Funding:** This work was supported by Hebei Provincial Health Commission Fund (Grant No. 20210302) and Hebei College of Traditional Chinese Medicine Fund (Grant No. KTY2019015).

**Competing interests:** The authors have declared that no competing interests exist.

## Conclusion

We screened the feature genes, CCL18, MMP10 and WIF1, among the differentially expressed genes (DEGs) by three different machine learning algorithms. They showed good diagnostic performance in both training and testing cohorts and were potential diagnostic markers for IC. We paint the immune landscape of IC. In IC tissue, immune cell subtypes infiltrated extensively. Most immune cell subtypes were up-regulated in IC tissue, including mast cells, activated CD4 T cells, and regulatory T cells that suppress immune responses. MMP10 and CCL18 had positive correlation to ICI, while WIF was negatively correlated with ICI. MMP10 and CCL18 may be the driving factors of immune response or their expression levels may be increased by immune response. The effect of characteristic genes of IC on immune cell subtypes still needed to be further explored.

## Introduction

Interstitial cystitis (IC) is a disease of unknown etiology characterized by frequent urination, urgency, and pelvic pain [1]. The lack of effective treatment adds to the confusion of patients with IC, which may bring negative emotion to patients, impair ability to participate in social activities, deprive patients of sleep time, and greatly reduce the quality of life of patients [2–4].

The etiology of IC is unknown which may be caused by multiple factors. IC was divided into ulcerative and non-ulcerative phenotypes depending on whether it is accompanied by hunner ulcer under cystoscope. Previous studies suggested several hypotheses about the cause of IC [5–18]. Inflammation may be an important cause of IC [5–9]. Inflammation was the main feature of ulcerative IC, which was characterized by extensive submucosal inflammation and perineural plasma cell and lymphocyte infiltration [8]. However, the inflammatory cell infiltration of non-ulcerative IC was less than those in ulcerative IC [5,6]. Therefore, whether IC is related to inflammation was worth being explored. IC was suggested to be associated with bacterial infections [10–12]. But so far no microbiological evidence has been found. It had been suggested that, as helicobacter pylori caused ulcers in the digestive tract, the ulcerative IC may be related to some bacteria that were currently difficult to culture and had not been discovered [10]. *Oravisto et al*. proposed that IC had a certain relationship with autoimmune diseases [13]. There were differences in the immune mechanisms involved in the development of IC subtypes [8]. The infiltration of T cells and B cells in the bladder submucosa of ulcerative IC were higher than that of non-ulcerative IC [8]. Mast cells, as a kind of multifunctional immune cells, were enriched in IC, which contain a large number of inflammatory mediators, such as histamine, leucotrienes, 5-HT and cytokines [14,15]. Mast cells in detrusor tissues of non-ulcerative IC were normal or only mildly elevated [16,17]. The previous study proposed the theory of urothelial dysfunction/glycosaminoglycan (GAG) defect [18]. IC patients have significantly increased levels of antiproliferative factors in their urine, which leads to the destruction of the GAG layer [18].

In addition to the unknown cause, IC was still a disease with the exclusive diagnosis and lacked an effective gold standard. Some previous studies explored the diagnostic markers of IC, including inflammatory mediators secreted by mast cells, inflammatory cell metabolites, chemokines, growth factors, and antiproliferative factors [19–24]. Due to lack of consensus on the diagnostic criteria IC, it was of great significance to find biomarkers for IC.

In our study, we applied machine learning algorithms to screening characteristic genes of the IC and validated the diagnostic efficacy in training and test cohorts. We paint an immune landcape, showing correlations between characteristic genes and immune cell subtypes. Our study identified potential diagnostic markers for IC and provides possible clues to the mechanistic elucidation of disease occurrence and progression.

## Methods

### Data sets download

RNA sequencing data sets of IC were downloaded from Gene Expression Omnibus (GEO; https://www.ncbi.nlm.nih.gov/geo/) database, including GSE57560, GSE11783 and GSE621, whose corresponding platforms were GPL16699, GPL570 and GPL262 respectively. GSE57560 include 13 IC tissues and 3 normal baldder tissues. GSE11783 included 5 IC tissues and 6 normal baldders. GSE621 contained 6 IC tissues and 6 normal baldder tissues (S1 Table). We took GSE57560 and GSE11783 as training cohort, and GSE621 as test cohort. All samples were batch normalized by R software (V4.0.2) and R package (model.matrix; ComBat) to remove the influence of different experimental conditions and environments on the results.

### Difference analysis and enrichment analysis

R software and R package (limma; sva) were used to integrate data sets (GSE57560 and GSE11783) and identify differentially expressed genes (DEGs). $|\log_2 \text{fold change}| > 1$ and P-value $< 0.05$ were set as the filter criterion for DEGs. Kyoto Encyclopedia of Genes and Genomes (KEGG) pathway enrichment analysis and Gene Ontology (GO) enrichment analysis were performed via R software and R package (clusterProfiler). False discovery rate (FDR) $<0.05$ was regarded as the filter criterion for GO items and KEGG pathway. R software and R packages (DOSE; enrichplot; Org.hs.e.g., db) were applied to conduct gene set enrichment analysis (GSEA) for DEGs. FDR $< 0.05$ was used as the screening standard.

### Machine learning algorithms screen key genes

Weighted gene co-expression network analysis (WGCNA) of all genes was constructed using R software and R package (WGCNA). We performed the Pearson's correlation matrices and average linkage method for all pairwise genes. Power function was applied to construct weighted adjacency matrix. The formula was $A_{mn} = |C_{mn}|^\beta$. $C_{mn}$ represented pearson correlation coefficient between gene m and n. $A_{mn}$ represented adjacency between gene m and gene n. β as a soft threshold parameter emphasized strong correlations between genes and penalize weak correlations. According to the scale-free fitting index $R^2$, which was 0.9 for the first time, the optimal soft threshold β was determined. The adjacency was transformed into a topological overlap matrix (TOM) and the corresponding dissimilarity (1-TOM) was calculated. Average linkage hierarchical clustering was conducted based on the TOM-based dissimilarity. The minimum number of genes of modules was set to 30, and the clipping height was set to 0.25 to combined similar module. The correlation between modules and clinical features was calculated. $|\text{Cor}| > 0.4$ and P-value $< 0.05$ were identified as filter criteria to retain modules, considered relatively significant. If the correlation coefficient is too low, it indicates that the importance of the module is minimal. Module Membership (MM) and gene significance (GS) were calculated in retained modules.

LASSO analysis was performed on all genes using R software and R package (glment). As a compression estimation method, the principle of Lasso regression analysis was to construct a penalty function, compressing some regression coefficients. The sum of the absolute values of the forced coefficients is less than a certain fixed value, and some regression coefficients were set to zero. The degree of complexity adjustment of LASSO regression was controlled by the

parameter λ. The larger the λ was, the greater the punishment was for the linear model. We determine lambda when the binomial deviance was the smallest. Genes that contribute little to the whole population were eliminated.

Using R software and R packages (Kernlab; Caret) performed support vector machine (SVM) operation on all genes. SVM was a supervised pattern recognition method. Its main idea was to build a classification decision surface. SVM uses kernel functions to map data into high-dimensional space, making it as linearly separable as possible. Radial basis functionl (RBF) showed good classification performance. We choose RBF as the classification kernel function of SVM. When cross-validation root mean square error (RMSE) was minimum, we determine the penalty coefficient C and the kernel parameter gamma to prevent overfitting.

### Immune cell infiltration calculation

Based on gene expression, single sample Gene Set Enrichment Analysis (ssGSEA) was used to calculate the ICI in all samples. This process was implemented with R software and R packages (GSVA). The ssGSEA algorithm is a method utilized for gene set enrichment analysis, primarily aimed at assessing the enrichment level of a specific gene set within an individual sample. ssGSEA computes the enrichment score for each gene set by comparing the gene expression matrix with predefined gene sets. Immune score and stromal score was figured out by ESTIMATE algorithm. Immune score indicated the abundance of immune cells in the tissue. The ESTIMATE algorithm serves as a quantitative method to estimate the levels of stromal and immune cell infiltration within the tumor microenvironment, leveraging gene expression data from cancer tissues. This algorithm predicts the levels of stromal and immune cell infiltration in tumor tissues through a pre-selected set of stromal/immune-related genes and generates three principal scores: Stromal Score, Immune Score, and Estimate Score. The Stromal Score assesses the level of stromal cell infiltration in tumor tissues, the Immune Score evaluates the level of immune cell infiltration, while the Estimate Score, being the sum of the two, can be used to infer tumor purity.

### Statistical analysis

Differences between groups were analyzed using Kruskal-Wallis test. Correlation analysis was conducted via Spearman test. Receiver operator characteristic curve (ROC) was conducted to evaluate diagnostic efficacy of key genes. P-value <0.05 on both sides indicated statistical significance. The data sets obtained were open source with no require of ethics committee review

## Results

### DEGs screening and enrichment analysis

There were 891 DEGs identified between IC and normal bladder tissue samples. Compared with normal bladder tissue samples, 432 genes were down-regulated and 459 genes were up-regulated in IC (S2 Table). The heatmap showed the top 50 genes with the largest |log$_2$FC| (Fig 1A).

The DEGs were enriched in the biological process (BP) items, which included leukocyte migration, leukocyte chemotaxis, leukocyte cell-cell adhesion, T cell activation and myeloid leukocyte migration. The DEGs were enriched in the cellular component (CC) items, which included external side of plasma membrane, tertiary granule, secretory granule membrane, cytoplasmic vesicle lumen, vesicle lumen, and secretory granule lumen. The DEGs were enriched in the molecular function (MF) items, which included immune receptor activity, cytokine activity, chemokine activity, chemokine receptor binding and receptor ligand activity. The bar graph showed the top five items with the lowest FDR in BP, CC, and MF (Fig 1B). DEGs were enriched in KEGG pathway related to immune cells migration (Fig 1C). The bar graph showed the top ten items with the lowest FDR in KEGG pathway. GSEA analysis identified important gene sets for normal group and IC group which were associated with immune response (Fig 1D and Fig 1E).

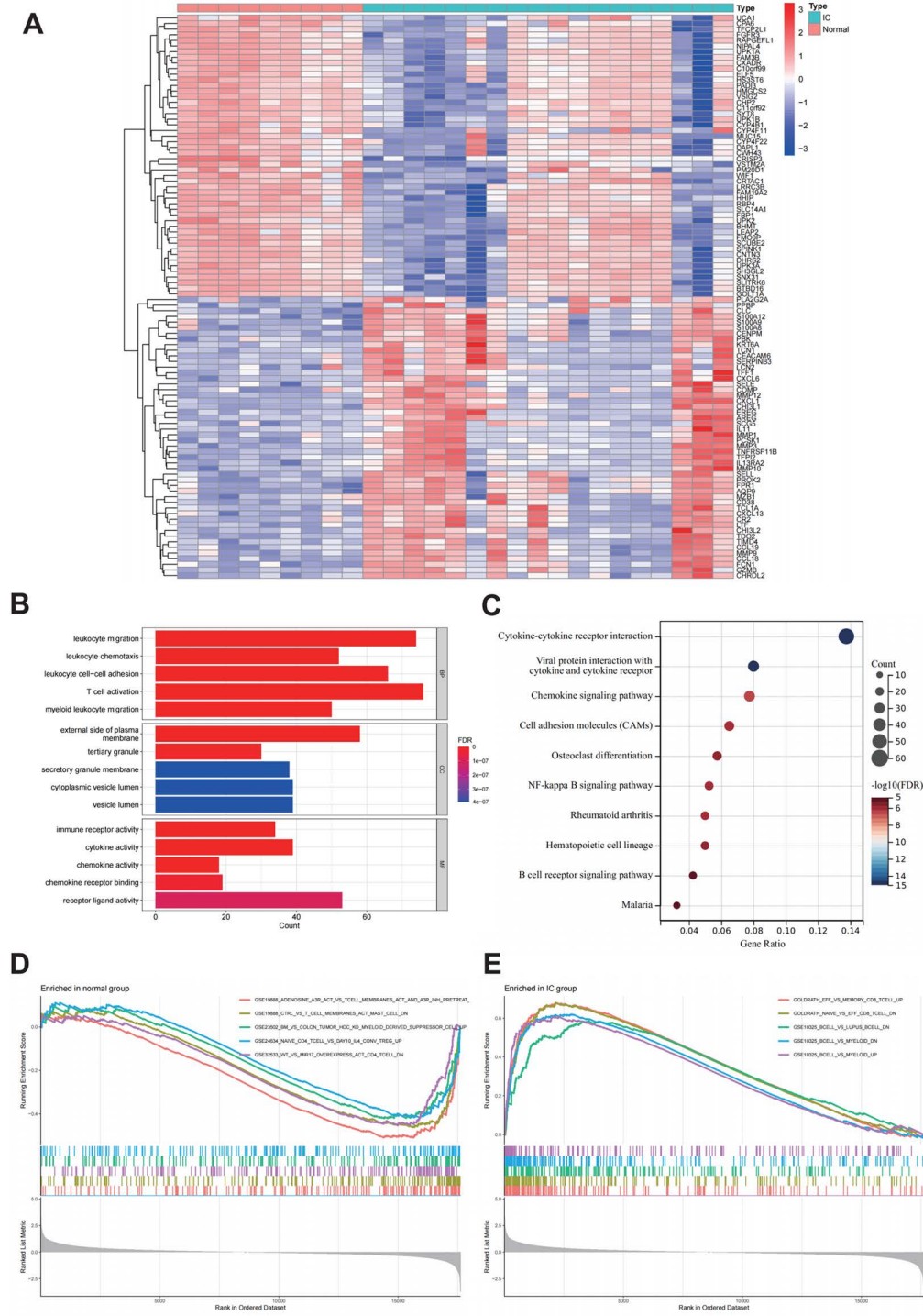

**Fig 1. A. Heatmap showing the expression levels of the top 50 genes with the largest absolute log2 fold change (|log2FC|).** Red color indicates upregulation, while blue color indicates downregulation.B: Gene Ontology (GO) enrichment analysis of the differentially expressed genes (DEGs). The x-axis represents the number of genes enriched in each GO term, the y-axis represents the GO terms, and the color represents the false discovery rate (FDR).C: Kyoto Encyclopedia of Genes and Genomes (KEGG) enrichment analysis of the DEGs. The x-axis represents the gene ratio, the y-axis represents the pathways, and the color represents the FDR.D and E: Gene Set Enrichment Analysis (GSEA) enrichment analysis in the normal group and interstitial cystitis (IC) group. Different colors represent different sets of genes.

 

## Key genes screeing by machine learning algorithms

According to the scale-free fitting index $R^2$, which was 0.9 for the first time, the optimal soft threshold was determined as $\beta = 19$ (Fig 2A). At this time, the mean connectivity of the network was relatively high and can contain enough information (Fig 2B). The minimum number of genes of modules was set to 30, and the clipping height was set to 0.25. A total of eight modules were obtained (Fig 2C). Module eigengene (ME) of blue module (r = -0.52, P = 0.006) were negatively correlated with IC, while ME of cyan module (r = 0.46, P = 0.02), pink module (r = 0.44, P = 0.02) was positively correlated with IC (Fig 2D). In blue (cor = 0.61, P = 5e-124), cyan (cor = 0.62, P = 1.4e-12) and pink (cor = 0.65, P = 4.4e-170) modules, MM was correlated to GS (Fig 2E-2G). Hub genes were identified by the criteria (MM > 0.8 and GS > 0.5). In the blue (S3 Table), cyan (S4 Table) and pink (S5 Table) modules, we obtained 232, 69 and 148 hub genes, respectively. Log(λ) was determined at the minimal binomial deviance value. Then, the key genes (n = 32) were selected by LASSO (Fig 3A). Key genes (n = 43) were selected with minimal cross validation RMSE by SVM algorithm (Fig 3B). The blue module, DEGs and the gene sets obtained by the other two algorithms had two overlapping genes, CCL18 and MMP10 (Fig 3C). The cyan and pink modules, DEGs and the gene sets obtained by the other two algorithms had one overlapping gene, WIF1 (Fig 3D).

## Expression level and diagnostic efficacy of key genes in training cohort and test cohort

In the training cohort, MMP10 (p < 0.001) and CCL18 (p < 0.01) were higher expressed in IC tissues compared with normal bladder tissues, while WIF1 (p < 0.001) was lower expressed in IC tissues (Fig 4A-4C). These differences were consistent across the training and test cohorts (Fig 4D-4F). In the training cohort, MMP10 (AUC = 0.938, 95 CI%(0.796, 1.000), p < 0.05), CCL18 (AUC = 0.944, 95 CI%(0.840, 1.000), p < 0.05), WIF1 (AUC = 0.883, 95 CI%(0.728, 0.981), p < 0.05) had good diagnostic efficacy for IC (Fig 4G-4I). In the test cohort, MMP10 (AUC = 0.920, 95 CI%(0.790, 1.000), p < 0.05), CCL18 (AUC = 0.864, 95 CI%(0.710, 0.975), p < 0.05), WIF1 (AUC = 0.784, 95 CI%(0.586, 0.932), p < 0.05) also had good diagnostic efficacy for IC (Fig 4J-4L).

## The relationship between key genes expression and ICI

Compared with normal bladder tissues, 18 immune cell subtypes were up-regulated in interstitial cystitis tissues (P < 0.05). The infiltration levels of the remaining 10 immune cell subtypes between normal bladder tissues and IC tissues was not statistically significant (P > 0.05, Fig 5A and Fig 5B). The infiltration level of most immune cell subtypes in IC were positively correlated, while CD56 + natural killer cells was negatively correlated to most other immune cell subtypes (P < 0.05, Fig 5C). Immune score (P < 0.05), stromal score (P < 0.05) and ESTIMATE score (P < 0.05) were higher in IC samples than in normal bladder samples (Fig 5D). The expression levels of CCL18 and MMP10 were positively correlated with most ICI in IC. Conversely, WIF1 expression was negatively correlated with most ICI in IC (P < 0.05, Fig 5E). MMP10 (r = 0.60, P = 0.011) and CCL18 (r = 0.89, P < 0.001) were positively correlated to immune scores in IC, while WIF1 (r = -0.60, P = 0.010)was negatively correlated to immune scores (Fig 5F-5H).

## Discussion

The clinical diagnosis of IC was mainly via cystoscopy and bladder biopsy after bladder water dilatation with isotonic salt solution [25–27]. Microscopically, the majority of IC patients had globular hemorrhagein the bladder [28]. Hunner ulcer were observed in only about 5%-15% of IC patients, which can also occur in other bladder diseases, even in a normal bladder [29]. The diagnosis method was not very not specific. Since there was still no consensus on the diagnostic criteria of IC, it was of great significance to find biomarkers for IC. In previous studies, several diagnostic markers had been developed. Mast cells were considered to be the characteristic immune cell subtype of IC [19]. The large amount of IL-6 secreted by mast cells had high sensitivity and specificity for the diagnosis of IC [20]. Inflammatory cell metabolites in the urine were potential diagnostic markers for IC. *Wen et al*. depicted the urine metabolic profile of IC based metabolomic analysis [21]. Urine metabolomics analysis from IC patients revealed 140 nuclear magnetic resonance peaks were

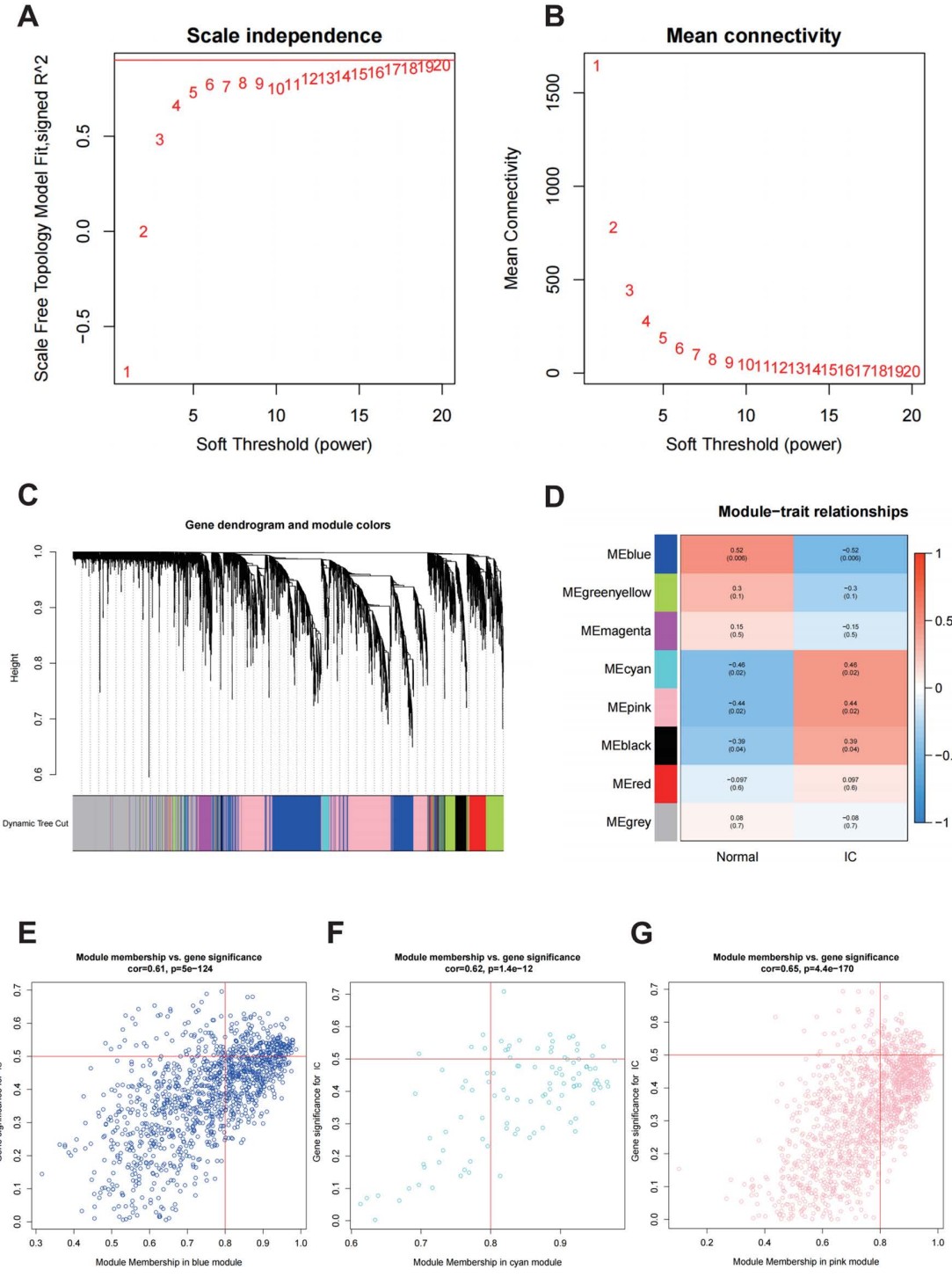

**Fig 2. Characteristics of gene screening by weighted gene co-expression network analysis (WGCNA).** A and B: Power value screening in the interstitial cystitis (IC) gene co-expression module. C: Gene dendrogram and co-expression module construction of IC. The tree branches represent genes, and the colors represent co-expression modules. D: Relationships between modules and clinical features. Red represents positive correlation, blue represents negative correlation. The number at the top of the square represents the correlation coefficient, and the number in parentheses represents the P-value. E-G: Hub genes screening in the blue module, cyan module, and pink module.

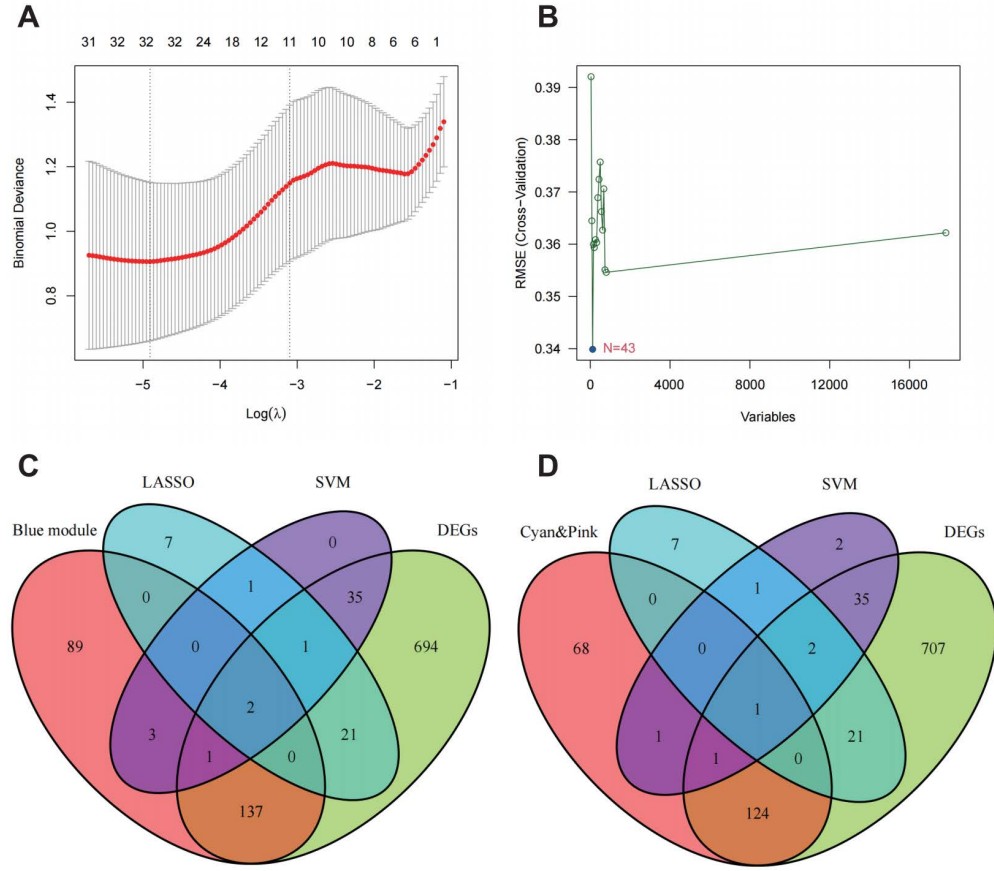

**Fig 3. Characteristics of gene screening by LASSO and SVM algorithms.** A: Characteristics of gene screening by LASSO algorithm. B: Characteristics of gene screening by SVM algorithm. C: Gene intersection from blue module, LASSO gene set, SVM gene set, and DEGs.D: Gene intersection from cyan and pink module, LASSO gene set, SVM gene set, and DEGs.

significantly up-regulated compared with control group. In IC patients, metabolites with the strongest magnetic resonance peak were tyramine (a kind of pain related to neural regulator) and 2- keto glutaric acid (the key to the Krebs cycle intermediates).

In our study, the DEGs were identified between IC tissue samples and normal bladder tissue samples based on RNA sequencing data set downloaded from GEO database. These DEGs were significantly enriched in GO items and KEGG pathways which were related to inflammation and immunity. DEGs were enriched in BP items, including leukocyte migration, leukocyte chemotaxis, leukocyte cell-cell adhesion, T cell activation, myeloid leukocyte migration, etc. These were consistent with previous pathological findings observed in IC tissues. Inflammatory cells, including neutrophils, T cells, B cells, and mast cells, were extensively infiltrated in the submucosa of IC tissue, especially in ulcerative IC [8]. DEGs were enriched in CC such as external side of plasma membrane, tertiary granule, secretory granule membrane, cytoplasmic vesicle lumen, vesicle lumen, secretory granule lumen. These were related to the secretory function of cells. DEGs were enriched in MF items, including immune receptor activity, cytokine activity, chemokine activity, chemokine receptor binding, receptor ligand activity, etc. These suggested that the occurrence of IC may be related to the activation of immune system. Under the action of chemokines, immune cells were enriched to the lesion. The enrichment of DEGs in KEGG pathway was also related to chemokine release and interaction. GO and KEGG enrichment analysis showed that the

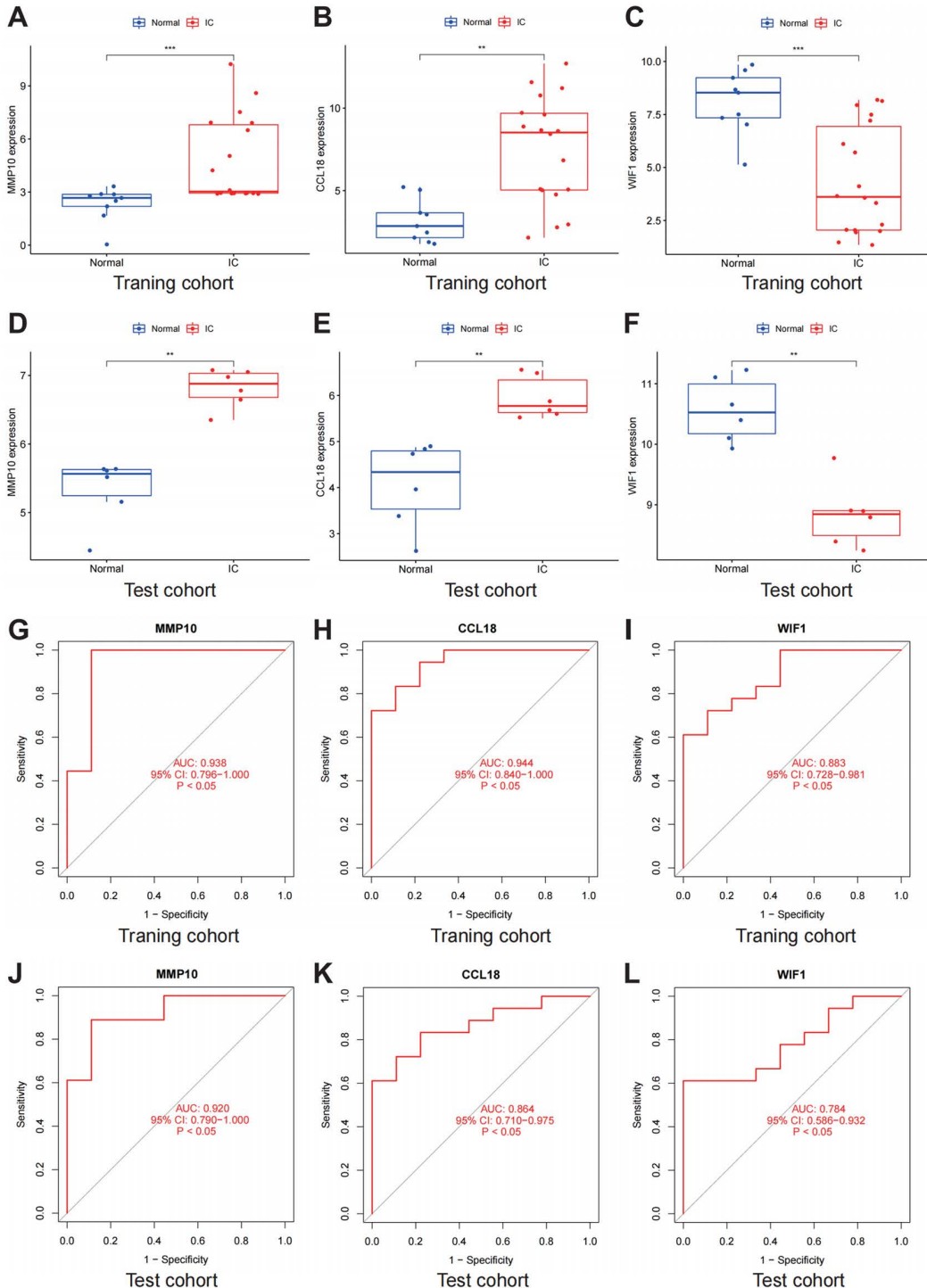

**Fig 4. Diagnostic efficiency of characteristic genes.** A-C: Comparison of MMP10, CCL18, and WIF1 expression between the interstitial cystitis (IC) and normal groups in the training cohort. D-F: Comparison of MMP10, CCL18, and WIF1 expression between the IC and normal groups in the test cohort. G-I: Receiver operator characteristic (ROC) curves of MMP10, CCL18, and WIF1 in the training cohort. J-L: ROC curves of MMP10, CCL18, and WIF1 in the test cohort. *represents p-value < 0.05, **represents p-value < 0.01, ***represents p-value < 0.001.

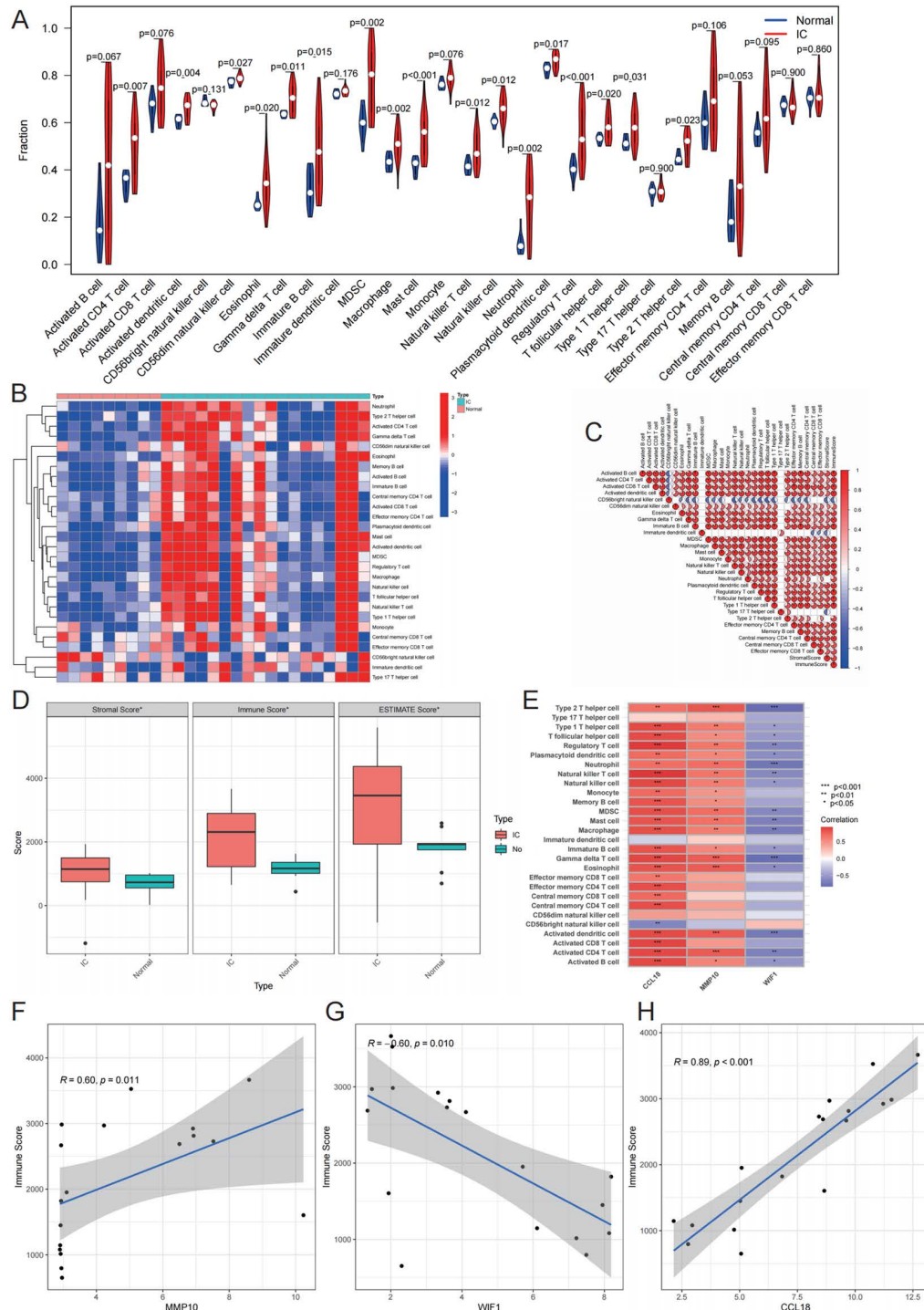

**Fig 5. Immune landscape of interstitial cystitis (IC).** A) Differences in immune cell infiltration (ICI) between IC samples and normal samples. B) Clustering heatmap showing the upregulation (red) and downregulation (blue) of ICI. C) Correlation between immune cell subtypes, with colors indicating the correlation coefficient. D) Differences in stromal score, immune score, and ESTIMATE score between IC samples and normal samples. E) Correlation between the expression of characteristic genes and ICI, with colors indicating the correlation coefficient. F-H) The relationship between MMP10, WIF1, CCL18, and immune score. *represents p-value < 0.05, **represents p-value < 0.01, ***represents p-value < 0.001.

occurrence of IC was closely related to inflammation and immune system activation, which provided clues for elucidating the pathogenesis of the disease.

Three machine learning algorithms were used to screen the characteristic genes of IC. There were three common genes in the gene sets obtained by different machine learning algorithms,namely CCL18, MMP10 and WIF1. All three machine learning algorithms have their own strengths and weaknesses, but they can complement each other's weaknesses. WGCNA uses the information of all genes to identify gene sets of interest and conduct significant association analysis with phenotypes. The advantage of this method is that it makes full use of the information and converts the association between genes and phenotypes into the association between gene sets and phenotypes, thus eliminating the problem of multiple hypothesis testing and correction. Sample heterogeneity will affect module identification. Different data preprocessing and analysis parameter selection will also lead to different results, such as gene expression normalization, correlation coefficient calculation, clustering, etc. The larger the sample size, the better the result. Lsaso regression can obtain a more streamlined model by constructing penalty functions to compress regression coefficients of some variables. Lasso regression analysis has relatively low requirements on data, and can be widely used in both continuous and discrete variables. However, the Lasso regression analysis ignore the group association, and the results are unstable. SVM solves machine learning in the case of small samples.Because the kernel function method is used to overcome the problem of dimensionality disaster and nonlinear divisibility, the computational complexity is not increased when mapping to higher dimensional space. The defect of SVM is that it is not suitable for large sample analysis and multi-classification, and partial sample information is lost. The combined application of these three algorithms can enhance the credibility of the results and the strength of evidence support, and solve the problem of small sample size and sample heterogeneity to a certain extent. We selected the overlapping genes from three algorithms as key genes and validated the reliability of the results through differential analysis and ROC curve assessment. The combined application of the three machine learning algorithms effectively reduces overfitting, enhances model performance, and strengthens robustness.

Compared with normal bladder tissues samples, CCL18 and MMP10 were highly expressed in IC tissues samples, and WIF1 was low expressed. These three genes had good diagnostic efficacy for IC. We obtained consistent results in the training and test cohort. The relationship between CCL18, MMP10, WIF and IC has not been elucidated in previous studies. CCL18 is a typical chemokine involved in immune regulation [30]. CCL18 induced the movement of immature T cells towards dendritic cells and activated macrophages in lymph nodes [31,32]. CCL18 was included in the migration of B cells to follicular cells. With effetcs on chemotactic activity of immature T cells, CD4[+] and CD8[+] T cells, CCL18 was vital to immune responses [33–35]. CCL18 was considered as a marker of macrophages M2 [36], and its high expression predicted poor prognosis for tumor patients [37,38], which may be related to immune escape of tumor cells and immunosuppression of tumor microenvironment. CCL18 also promoted epithelial-mesenchymal transition and angiogenesis [39]. Metalloproteinases (MMP) was a key part in the regulation of inflammatory reaction, tumor, tissue and organ remodeling and other physiological processes. The expression of MMP10 was low or not at all in tissues under normal conditions. However, when the body was exposed to external stimuli, including injury and infection, the expression of MMP10 significantly increased [40,41]. MMP10 was highly expressed in the infiltrating macrophages. The previous study pointed out that the up-regulation of MMP10 expression was accompanied by down-regulation of CCL2 expression [40]. The upregulation of CCL2 expression predicted macrophage activation. MMP10 may inhibit macrophage activation and play an anti-inflammatory role [41]. A previous study suggested that MMP10 may cause kidney podocyte damage and accelerate the process of kidney sclerosis [42]. WIF1 was regard as an inhibit factor of the Wnt signaling pathway [43,44], which may regulate the differentiation of Th17 cells, the activation of cytotoxic T cells, the immune tolerance of dendritic cell, the formation of CD8[+] memory T cells, the proliferation and function of regulatory T cells [45–48]. This study differs from previous research in that the biomarkers of IC we have identified have not been reported in the existing literature, making them worthy of further exploration.

Based on GSEA, We discovered that 18 immune cell subtypes infiltration were higher in IC tissues than that in normal bladder tissues, which included mast cells, macrophages, activated CD4 T cells, myeloid-derived Suppressor cells (MDSC), regulatory T cells, and T follicular helpers Cell, type 1 T helper cell, and type 2 T helper cell, etc. Previous study showed that mast cells were activated and aggregated in IC under the joint action of a variety of related factors, which are localized in the submucosal and muscularis of the bladder [19]. Many manifestations of ulcerative IC, such as pain, frequent urination, edema, local fibrosis and neovascularization, are closely related to mast cells [15]. The number of submucosal mast cells in ulcerated IC was 10 fold higher than that in the control group [19]. Therefore, mast cells are not only an important part in the pathogenesis of ulcerated IC, but also a specific marker. Some studies defined the pathogenesis of IC as the "mast cell hypothesis", but the detailed pathogenesis of IC remains unclear [15,18,29]. We found that cells with negatively regulate immune function, such as MDSC and regulatory T cells, were highly expressed in IC. Activation and proliferation of these cells negatively regulated inflammatory response, which was a protective mechanism of the body. The immune score can represent the content of all immune cell subtypes in tissues. ICI and stromal cells infiltration in IC tissues were higher than that in normal samples, and normal urothelial cells were reduced. These results suggested that the occurrence of IC may be related to the activation, proliferation and enrichment of immune cells. CCL18 and MMP10 were positively correlated with most immune cell subtypes and immune score. This may suggest that CCL18 and MMP10 were the drivers of immune responses in IC. WIF1 was negatively correlated with immune score and most immune cell subtypes. WIF may inhibit the proliferation, activation and migration of immune cells by Wnt pathway. The role of characteristic genes in immune responses needed to be further explored. The roles of mast cells, regulatory T cells, and macrophages in IC have been well established. Several immune-related targets are currently under close scrutiny. Potential therapeutic strategies may include the inhibition of various cytokines, chemokines, growth factors, and mast cells. Certolizumab pegol and adalimumab have undergone clinical evaluation. Tanezumab and fulranumab have become subjects of research. Gene therapies with immunomodulatory properties may play a significant role in addressing diseases such as IC, although further exploration is still required.

The biggest drawback of this study was the small sample size included in the study, and we can't get more samples, which is a great pity. This was related to the fact that IC has received less attention. Sample heterogeneity can also have a disastrous effect on the stability of the results. Although we used three machine learning algorithms to improve the confidence of the results, we still believed that further expansion of the sample size for validation was needed to explore the role of key genes in disease occurrence and progression. We recommend addressing the shortcomings associated with the small sample size from the following perspectives: expanding the sample size by further collecting samples from interstitial bladder studies for comprehensive transcriptomic sequencing; conducting multi-omics integrative analyses to validate conclusions across different omics; and performing foundational experiments to elucidate the causal relationships between genes and disease. The inability to elucidate the role of key genes in interstitial cystitis represents another limitation of this study, which we have emphasized in the discussion section. There is no existing literature that clarifies the relationship between CCL18, MMP10, and WIF1 with interstitial cystitis. We were unable to find support from previous studies. However, our research may have expanded the direction of interstitial cystitis studies. CCL18, MMP10, and WIF1 could potentially serve as biological markers, warranting further exploration. In addition, a shortcoming of this study is the lack of basic experiments to verify the diagnostic efficacy and underlying mechanisms of key genes

## Conclusion

We screened the feature genes, CCL18, MMP10 and WIF1, among the DEGs by three different machine learning algorithms. They showed good diagnostic performance in both training and testing cohorts and were potential diagnostic markers for IC. We paint the immune landscape of IC. In IC tissue, immune cell subtypes infiltrated extensively. Most immune cell subtypes were up-regulated in IC tissue, including mast cells, activated CD4T cells, and regulatory T cells that suppress immune responses. MMP10 and CCL18 were positively correlated with ICI, while WIF was negatively correlated

with ICI. MMP10 and CCL18 may be the driving factors of immune response or their expression levels may be increased by immune response. The effect of characteristic genes on pathogenic mechanism of IC and the immune cell subtypes still needed to be further explored.

## Supporting information

**Table S1. The information of samples.**
(XLSX)

**Table S2. Differentially expressed genes identified between IC and normal bladder tissue samples.**
(XLSX)

**Table S3. Hub genes in the blue module.**
(XLSX)

**Table S4. Hub genes in the cyan module.**
(XLSX)

**Table S5. Hub genes in the pink module.**
(XLSX)

## Author contributions

**Data curation:** Jie Hu, Jianpeng Han, Jianyong Feng, Kuo Guo, Wenbin Chen.

**Project administration:** Yongzhang Li.

**Writing – original draft:** Junhao Wang, Yan Zhou.

**Writing – review & editing:** Junhao Wang, Yan Zhou, Yanrui Yun.

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
