## [Decision Letter · Decision Letter 0]

27 Dec 2024

Dear Dr. Li,

Thank you for submitting your manuscript to PLOS ONE. After careful consideration, we feel that it has merit but does not fully meet PLOS ONE’s publication criteria as it currently stands. Therefore, we invite you to submit a revised version of the manuscript that addresses the points raised during the review process.

We look forward to receiving your revised manuscript.

Kind regards,

Abdul Rauf Shakoori, PhD

Academic Editor

PLOS ONE

Journal requirements: When submitting your revision, we need you to address these additional requirements. 1. Please ensure that your manuscript meets PLOS ONE's style requirements, including those for file naming. The PLOS ONE style templates can be found at https://journals.plos.org/plosone/s/file?id=wjVg/PLOSOne_formatting_sample_main_body.pdf and https://journals.plos.org/plosone/s/file?id=ba62/PLOSOne_formatting_sample_title_authors_affiliations.pdf. 2. PLOS requires an ORCID iD for the corresponding author in Editorial Manager on papers submitted after December 6th, 2016. Please ensure that you have an ORCID iD and that it is validated in Editorial Manager. To do this, go to ‘Update my Information’ (in the upper left-hand corner of the main menu), and click on the Fetch/Validate link next to the ORCID field. This will take you to the ORCID site and allow you to create a new iD or authenticate a pre-existing iD in Editorial Manager. 3. Please note that PLOS ONE has specific guidelines on code sharing for submissions in which author-generated code underpins the findings in the manuscript. In these cases, we expect all author-generated code to be made available without restrictions upon publication of the work. Please review our guidelines at https://journals.plos.org/plosone/s/materials-and-software-sharing#loc-sharing-code and ensure that your code is shared in a way that follows best practice and facilitates reproducibility and reuse.  4. We note that the grant information you provided in the ‘Funding Information’ and ‘Financial Disclosure’ sections do not match.  When you resubmit, please ensure that you provide the correct grant numbers for the awards you received for your study in the ‘Funding Information’ section.

Reviewers' comments:

Reviewer's Responses to Questions

**Comments to the Author**

1. Is the manuscript technically sound, and do the data support the conclusions?

Reviewer #1: Yes

2. Has the statistical analysis been performed appropriately and rigorously?

Reviewer #1: Yes

3. Have the authors made all data underlying the findings in their manuscript fully available?

Reviewer #1: No

4. Is the manuscript presented in an intelligible fashion and written in standard English?

Reviewer #1: No

Reviewer #1: 1. Experimental Validation: While the manuscript successfully identifies three characteristic genes (CCL18, MMP10, and WIF1) for interstitial cystitis (IC) using machine learning, there is a lack of experimental validation. Experimental methods, such as qPCR, Western blotting, or immunohistochemistry, could confirm these genes' differential expression and diagnostic utility. Including at least one validation step would significantly strengthen the study.

2. Functional Analysis of Key Genes: The discussion regarding the biological relevance of CCL18, MMP10, and WIF1 is comprehensive. However, their functional roles in IC pathogenesis remain speculative. Please provide a deeper exploration of how these genes contribute to IC's immune landscape and their interaction with upregulated immune cells, supported by experimental or literature evidence.

3. Immune Landscape Analysis: The manuscript highlights the immune landscape of IC, with specific immune cell subtypes showing differential infiltration. The role of mast cells, regulatory T cells, and macrophages is well-documented in IC. However, linking these findings to therapeutic implications or pathways (e.g., targeting immune checkpoints or cytokines) would enhance the clinical impact of this analysis.

4. Small Sample Size: The study's conclusions are limited by the small sample size, as acknowledged by the authors. It would be valuable to discuss how this limitation might affect the reliability of the identified biomarkers and immune cell correlations. Consider suggesting future directions for validating findings with larger datasets or independent cohorts.

5. Machine Learning Models: The manuscript uses three machine learning algorithms (WGCNA, LASSO, SVM) to identify characteristic genes. While the multi-algorithm approach is commendable, the manuscript would benefit from explaining how the integration of these methods addresses potential biases or variability. Discuss the criteria for selecting overlapping genes and justify the robustness of the final results.

6. Figure Quality:

- Font Size: Many figures (e.g., heatmaps, ROC curves) have text that is too small and difficult to read. Increasing font size would improve clarity.

- Figure Legends: Ensure figure legends are detailed enough to be understood independently of the main text.

7. Clarity in Methods:

- The description of ssGSEA and ESTIMATE algorithms is concise but would benefit from a brief explanation for readers unfamiliar with these techniques.

- Specify the thresholds used in WGCNA and other machine learning algorithms, such as the rationale behind choosing |Cor| > 0.4.

8. Language and Style:

- Minor grammatical errors and awkward phrasing are present throughout the manuscript. For example, "depict the immune landcape" can be revised to "characterize the immune landscape."

- Consider a thorough language edit for improved readability and professionalism.

9. Literature Context:

- Add comparisons with previous studies that have identified biomarkers or immune cell dynamics in IC. Discuss how the current findings align or differ from these studies.

10. Supplementary Information:

- Ensure that the data and supplementary tables are adequately cross-referenced in the main text.

**Do you want your identity to be public for this peer review?** For information about this choice, including consent withdrawal, please see our Privacy Policy

Reviewer #1: No

---

## [Author Response · Author response to Decision Letter 0]

11 Feb 2025

Dear Editor,

We deeply appreciate your constructive comments on our manuscript (Title: Characteristic Genes and Immune Landscape of Interstitial Cystitis; ID: PONE-D-24-19821). We have made corrections according to the comments, and the detailed point-by-point answers to the questions are listed below. In the revised manuscript, the revised portions are marked with red font. Thank you very much. During the manuscript revision process, Professor Yanrui Yun provided extensive guidance. In recognition of her contributions to this research work, we have added her as the co corresponding author of the article.

Thank you once again for the suggestions and we hope that our revisions will meet with approval. We look forward to hearing from you.

Sincerely yours,

Yongzhang Li

Department of urology,

Hebei Provincial Hospital of Traditional Chinese medicine,

389 Zhongshan East Road, Shijiazhuang, Hebei Province, 050017 China

Email:hbzylyz@126.com

Phone: 08617532150046

Reviewer #1: 1. Experimental Validation: While the manuscript successfully identifies three characteristic genes (CCL18, MMP10, and WIF1) for interstitial cystitis (IC) using machine learning, there is a lack of experimental validation. Experimental methods, such as qPCR, Western blotting, or immunohistochemistry, could confirm these genes' differential expression and diagnostic utility. Including at least one validation step would significantly strengthen the study.

Re: Thank you for your comments. We agree with you. The lack of experimental verification is one of the biggest defects of this paper. We are collecting samples, which may take a while, and we are trying to do a continuity study. In our discussion, we highlighted our shortcomings.

2. Functional Analysis of Key Genes: The discussion regarding the biological relevance of CCL18, MMP10, and WIF1 is comprehensive. However, their functional roles in IC pathogenesis remain speculative. Please provide a deeper exploration of how these genes contribute to IC's immune landscape and their interaction with upregulated immune cells, supported by experimental or literature evidence.

Re: We sincerely appreciate your comments. The inability to elucidate the role of key genes in interstitial cystitis represents a limitation of this study, which we have emphasized in the discussion section. There is no existing literature that clarifies the relationship between CCL18, MMP10, and WIF1 with interstitial cystitis. We were unable to find support from previous studies. However, our research may have expanded the direction of interstitial cystitis studies. CCL18, MMP10, and WIF1 could potentially serve as biological markers, warranting further exploration.

3. Immune Landscape Analysis: The manuscript highlights the immune landscape of IC, with specific immune cell subtypes showing differential infiltration. The role of mast cells, regulatory T cells, and macrophages is well-documented in IC. However, linking these findings to therapeutic implications or pathways (e.g., targeting immune checkpoints or cytokines) would enhance the clinical impact of this analysis.

Re: Thank you very much for your comments. We have made supplementary additions during our discussion, which are as follows: The roles of mast cells, regulatory T cells, and macrophages in interstitial cystitis (IC) have been well established. Several immune-related targets are currently under close scrutiny. Potential therapeutic strategies may include the inhibition of various cytokines, chemokines, growth factors, and mast cells. Certolizumab pegol and adalimumab have undergone clinical evaluation. Tanezumab and fulranumab have become subjects of research. Gene therapies with immunomodulatory properties may play a significant role in addressing diseases such as IC, although further exploration is still required.

4. Small Sample Size: The study's conclusions are limited by the small sample size, as acknowledged by the authors. It would be valuable to discuss how this limitation might affect the reliability of the identified biomarkers and immune cell correlations. Consider suggesting future directions for validating findings with larger datasets or independent cohorts.

Re: Thank you for your comments. We concur with your perspective and have made additional clarifications in our discussion, which are as follows: The primary limitation of this study is the relatively small sample size, which precludes the acquisition of a more diverse set of samples, a situation that is indeed regrettable. This limitation is associated with the relatively low level of attention directed towards interstitial cystitis (IC). Additionally, the heterogeneity of the samples may have a catastrophic impact on the stability of the results. We recommend addressing the shortcomings associated with the small sample size from the following perspectives: expanding the sample size by further collecting samples from interstitial bladder studies for comprehensive transcriptomic sequencing; conducting multi-omics integrative analyses to validate conclusions across different omics; and performing foundational experiments to elucidate the causal relationships between genes and disease.

5. Machine Learning Models: The manuscript uses three machine learning algorithms (WGCNA, LASSO, SVM) to identify characteristic genes. While the multi-algorithm approach is commendable, the manuscript would benefit from explaining how the integration of these methods addresses potential biases or variability. Discuss the criteria for selecting overlapping genes and justify the robustness of the final results.

Re: Thank you for your feedback; we wholeheartedly agree with your perspective. We have supplemented our discussion as follows: We selected the overlapping genes from three algorithms as key genes and validated the reliability of the results through differential analysis and ROC curve assessment. The combined application of the three machine learning algorithms effectively reduces overfitting, enhances model performance, and strengthens robustness.

6. Figure Quality:

- Font Size: Many figures (e.g., heatmaps, ROC curves) have text that is too small and difficult to read. Increasing font size would improve clarity.

- Figure Legends: Ensure figure legends are detailed enough to be understood independently of the main text.

Re: Thank you very much for your feedback. We have adjusted the font size of the text in the images. We have also revised the legends to make them more comprehensive, allowing them to stand independently from the main text.

7. Clarity in Methods:

- The description of ssGSEA and ESTIMATE algorithms is concise but would benefit from a brief explanation for readers unfamiliar with these techniques.

- Specify the thresholds used in WGCNA and other machine learning algorithms, such as the rationale behind choosing |Cor| > 0.4.

Re: Thank you for your suggestions; we have made additional clarifications. The ssGSEA algorithm (Single Sample Gene Set Enrichment Analysis) is a method utilized for gene set enrichment analysis, primarily aimed at assessing the enrichment level of a specific gene set within an individual sample. ssGSEA computes the enrichment score for each gene set by comparing the gene expression matrix with predefined gene sets. The ESTIMATE algorithm (Estimation of Stromal and Immune cells in Malignant Tumor tissues using Expression data) serves as a quantitative method to estimate the levels of stromal and immune cell infiltration within the tumor microenvironment, leveraging gene expression data from cancer tissues. This algorithm predicts the levels of stromal and immune cell infiltration in tumor tissues through a pre-selected set of stromal/immune-related genes and generates three principal scores: Stromal Score, Immune Score, and Estimate Score. The Stromal Score assesses the level of stromal cell infiltration in tumor tissues, the Immune Score evaluates the level of immune cell infiltration, while the Estimate Score, being the sum of the two, can be used to infer tumor purity. In the context of WGCNA (Weighted Gene Co-expression Network Analysis), the criteria for retaining modules are determined by |Cor| > 0.4 and p-value < 0.05. A correlation of |Cor| > 0.4 is considered relatively significant. If the correlation coefficient is too low, it indicates that the importance of the module is minimal.

8. Language and Style:

- Minor grammatical errors and awkward phrasing are present throughout the manuscript. For example, "characterize the immune landscape" can be revised to "characterize the immune landscape."

- Consider a thorough language edit for improved readability and professionalism.

Re: Thank you for your comments. We have polished the language.

9. Literature Context:

- Add comparisons with previous studies that have identified biomarkers or immune cell dynamics in IC. Discuss how the current findings align or differ from these studies.

Reply: Thank you for your comments. We have supplemented the discussion as follows: This study differs from previous research in that the biomarkers we have identified have not been reported in the existing literature, making them worthy of further exploration.

10. Supplementary Information:

- Ensure that the data and supplementary tables are adequately cross-referenced in the main text.

Re: Thanks for your comments, we have provided supplementary tables.

---

## [Editor Report · Decision Letter 1]

17 Feb 2025

Characteristic Genes and Immune Landscape of Interstitial Cystitis.

PONE-D-24-19821R1

Dear Dr. Li,

We’re pleased to inform you that your manuscript has been judged scientifically suitable for publication and will be formally accepted for publication once it meets all outstanding technical requirements.

Kind regards,

Abdul Rauf Shakoori, PhD

Academic Editor

PLOS ONE
---

## [Editor Report · Acceptance letter]

PONE-D-24-19821R1

PLOS ONE

Dear Dr. Li,

I'm pleased to inform you that your manuscript has been deemed suitable for publication in PLOS ONE. Congratulations! Your manuscript is now being handed over to our production team.

Kind regards,

on behalf of

Prof. Dr. Abdul Rauf Shakoori

Academic Editor

PLOS ONE